analytical chemistry

terbinafine hydrochloride, resonance Rayleigh scattering, pharmaceutical dosage forms

**Author for correspondence:**
Sayed M. Derayea
e-mail: sayed.derayea@gmail.com

# Investigating the interaction of terbinafine with xanthenes dye for its feasible determination applying the resonance Rayleigh scattering technique

Mohamed A. Abdel-Lateef[1], Sayed M. Derayea[2], Deena A. M. Nour El-Deen[2], Albandary Almahri[3] and Mohamed Oraby[4]

[1]Department of Pharmaceutical Analytical Chemistry, Faculty of Pharmacy, Al-Azhar University, Assiut Branch, Assiut 71524, Egypt
[2]Department of Analytical Chemistry, Faculty of Pharmacy, Minia University, Minia 61519, Egypt
[3]General Courses Unit, Faculty of Sciences and Arts, King Khalid University, Dhahran Aljanoub, Saudi Arabia
[4]Department of Pharmaceutical Analytical Chemistry, Faculty of Pharmacy, Sohag University, Sohag 82524, Egypt

MAA-L, 0000-0002-3020-4966; SMD, 0000-0001-5357-1761

Terbinafine hydrochloride is a potent antifungal drug indicated for oral and topical treatment of mycoses. A resonance Rayleigh scattering (RRS) method was developed for the determination of terbinafine hydrochloride through a feasible complexation reaction with erythrosine B. In a weakly acidic medium (acetate buffer, pH 5.0), terbinafine hydrochloride can react with erythrosine B through the electrostatic attraction and virtue of hydrophobic force to form an ion-association complex. The reaction resulted in the appearance of a new RRS peak at 369 nm. The RRS peak was increased by increasing the concentration of terbinafine hydrochloride in the linear range of $0.1–1.5\ \mu g\ ml^{-1}$. All the reaction conditions (erythrosine B concentration, buffer volume, diluting solvent and pH) were optimized. The detection limit was $0.029\ \mu g\ ml^{-1}$ while the quantitation limit was $0.089\ \mu g\ ml^{-1}$. The suggested method after its validation was successfully applied for the determination of terbinafine hydrochloride in different pharmaceutical formulations (tablets and cream) with sufficient recovery.

**Figure 1.** Association complex formation between terbinafine and erythrosine B in the acidic medium.

# 1. Introduction

The last decades have seen unmatched alterations in the fungal infections pattern in individuals [1]. These fungal diseases have undertaken considerable importance owing to their rising incidence in patients with acquired immune deficiency syndrome disease, in recipients of solid organ transplants, in persons with malignancies diseases and in other immune-compromised individuals [1]. In addition, skin mycoses affect more than 25 per cent of the world's population, contributing one of the most frequent infection forms [2]. Terbinafine hydrochloride is an allylamine derivative (figure 1) that has a powerful antifungal activity through inhibition of the squalene epoxidase enzyme in fungal cell wall synthesis [3]. Therefore, terbinafine is applied for the treatment of cutaneous candidiasis, dermatophytoses, pityriasis versicolor and superficial fungal infections like onychomycosis, seborrheic dermatitis and tinea capatis particularly because of its short-duration therapy [4,5]. Terbinafine hydrochloride is pharmaceutically formulated as tablets for oral administration to treat toenail and fingernail fungus infection, or as a cream and powder formulations for superficial skin infections such as ringworm, athlete's foot (tinea pedis) and jock itch (tinea cruris). Terbinafine hydrochloride is a highly lipophilic drug in nature and tends to accumulate in fatty tissues, skin and nails [6]. Excessive terbinafine may produce some undesirable side effects like allergic reactions (swelling of tongue and face, difficulty in breathing and throat closing), skin rash, blood problems and changes in vision [4].

The analytical methods for the determination of terbinafine hydrochloride either in pharmaceutical dosage forms or in biological fluids were reviewed by Kanakapura & Penmatsa [4]. The reported spectrophotometric methods were based mainly on the ion-pair complex formation with some dyes (methyl orange, orange G, alizarin red S, bromothymol blue, bromophenol blue, bromocresol green and molybdenum (V) thiocyanate) followed by inconvenient extraction for the formed complexes with a suitable organic solvents [7–9]. In addition, the cited drug was determined by chromatographic techniques coupled with various detection patterns [10–15]. It is well known that chromatographic techniques require large volumes of hazardous organic solvents, tedious procedures, sometimes need expensive detectors and consume a long time. Fluorescence spectrometry is one of the analytical instrumentations that serves the purpose of high sensitivity without the loss of specificity or precision [16,17].

Green analytical chemistry can be defined as the using of techniques and the analytical methodologies that minimize or delete using or production of by-products, solvents, reagents, etc., that are toxic to human health or hazardous to the environment [18]. Recently, xanthene dyes such as eosin, and erythrosine B were

widely applied for the determination of many basic lipophilic drugs through spectrofluorometric [19,20], or resonance Rayleigh scattering (RRS) techniques [21,22] without organic solvent extraction. In this work, we aim to omit the organic solvent use during the analytical methodology through applying the RRS technique for the determination of terbinafine in different pharmaceutical dosage forms based on an ion-pair formation with erythrosine B dye.

# 2. Experimentation

## 2.1. Apparatus

All measurements of the RRS techniques were performed using a Scinco Fluorescence spectrometer equipped with 150 W Xe-arc lamp. The slit width for monochromators was set at 5 nm and photomultiplier tube voltage at 400 V.

## 2.2. Materials and reagent

Erythrosine B (Market Harborough Leicestershire, UK) was prepared in distilled water with a concentration of $2.27 \times 10^{-4}$ M, (200 µg ml$^{-1}$). The working standard solution of terbinafine hydrochloride (Egyptian Group for Pharmaceutical Industries, El-Obour city, Egypt) was prepared in double-distilled water at concentrations of 100 µg ml$^{-1}$. Acetate buffer solutions with various pH values were prepared by mixing 0.2 M of sodium acetate with 0.2 M of acetic acid in various proportions. Lamifen® 125 mg tablets and Lamifen® 1% cream (Egyptian Group for Pharmaceutical Industries, El-Obour city, Egypt) were obtained from the local market.

## 2.3. General analytical procedure

A suitable volume of terbinafine hydrochloride standard solution to give final concentrations in the range of (0.1–1.5 µg ml$^{-1}$), 1.0 ml of acetate buffer solution (pH 5.0) and 1.4 ml of erythrosine B solution (200 µg ml$^{-1}$) were mixed consecutively in a series of 10 ml volumetric flasks. The mixtures were completed to the calibration mark with double-distilled water and thoroughly mixed. The RRS spectra of the resulted solution was scanned by synchronous spectrofluorimetry (emission wavelength = excitation wavelength). The difference in the scattering intensity ($\Delta I_{RRS}$) was calculated at the maximum scattered wavelength of 369 nm as

$$\Delta I_{RRS} = I - I_\circ,$$

where $I$ is the scattering intensity for the formed ion-pair complex and $I_\circ$ is the scattering intensity for the blank.

## 2.4. Procedures for pharmaceutical tablets

Ten tablets of Lamifen® 125 mg were finely powdered. An accurately weighed amount of the resulting powder equivalent to 100 mg of terbinafine hydrochloride was transferred to a 100 ml standard flask and sonicated with about 30 ml of methanol for 5 min. The solution was completed up to the mark with the same solvent and filtered. Ten millilitres from the filtrate was quantitatively transferred to a 100 ml standard flask and completed up to the mark with distilled water. The final solution was analysed using the general procedure.

## 2.5. Procedures for pharmaceutical cream

An accurately weighed amount of Lamifen® 1% cream equal to 15 mg of terbinafine hydrochloride was quantitatively transferred into a clean dry 50 ml volumetric flask and 30 ml of methanol was added. The flask content was sonicated at 45°C for 30 min and completed to the mark with the same solvent. The mixture was cooled in an ice bath to solidify the base and filtered. An appropriate volume of the filtrate was diluted with distilled water to get a working solution within the recommended concentration range and analysed by the general procedure.

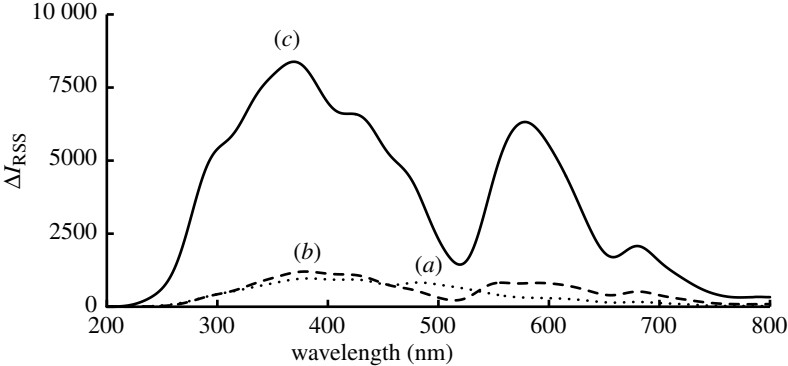

**Figure 2.** RRS spectra at pH 5.0: (*a*) for 1.0 μg ml⁻¹ terbinafine, (*b*) for erythrosine B (2.27 × 10⁻⁵ M), and (*c*) for the reaction product between erythrosine B and terbinafine (1.0 μg ml⁻¹).

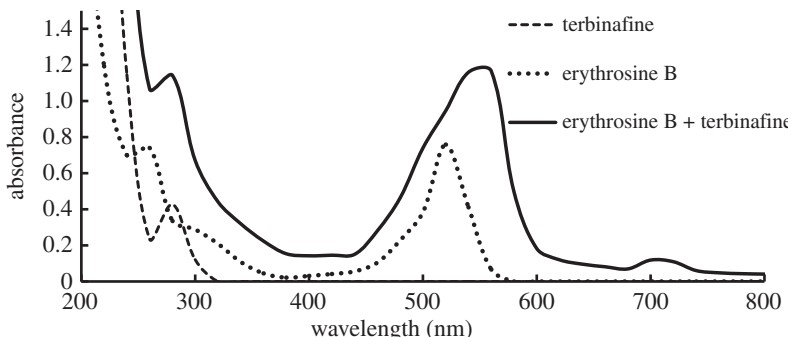

**Figure 3.** Absorption spectra of terbinafine, erythrosine B and their association complex.

# 3. Results and discussion

## 3.1. Resonance Rayleigh scattering spectrum

Terbinafine is a lipophilic drug ($\log_p = 6.0$) and contains one basic centre [6]. These features give the compound the ability to interact with xanthene dyes in the acidic medium [23]. The RRS spectrum of terbinafine−erythrosine B system is shown in figure 2. Both terbinafine and erythrosine B have very weak RRS intensities. Upon complex formation, the RRS was greatly enhanced with the appearance of three distinct RRS peaks at 369 nm, 570 nm and 681 nm. The intensity of all peaks has a linear relationship with terbinafine concentration, which can be applied for the determination of terbinafine hydrochloride. Because the sensitivity of the peak at 369 nm is much higher than that of the other two peaks, 369 nm was selected as the wavelength for the determination of terbinafine in the mentioned concentration range.

### 3.2.1. Relation between the absorption and resonance Raleigh scattering spectra

When the RRS peak is located at or near the molecular absorption band, the scattering process can absorb the light energy through resonance, resulting in a re-scattering process that produces a significant enhancement in the RRS intensity. Therefore, RRS spectra should be closely related to the absorption spectra [24,25]. By comparing the absorption spectrum (figure 3) with the RRS spectrum (figure 2), it was found that three RRS peaks at 369 nm, 570 nm and 680 nm closely correspond to absorption peaks at 300 nm, 550 nm and 720 nm, respectively. Thus, the RRS intensity was increased remarkably.

### 3.2.2. The molecular volume enlargement

According to the Rayleigh scattering formula, the RRS intensity is positively affected by the increase in the molecular scattering volume. Owing to the difficulty in calculating the molecular volume, it can be

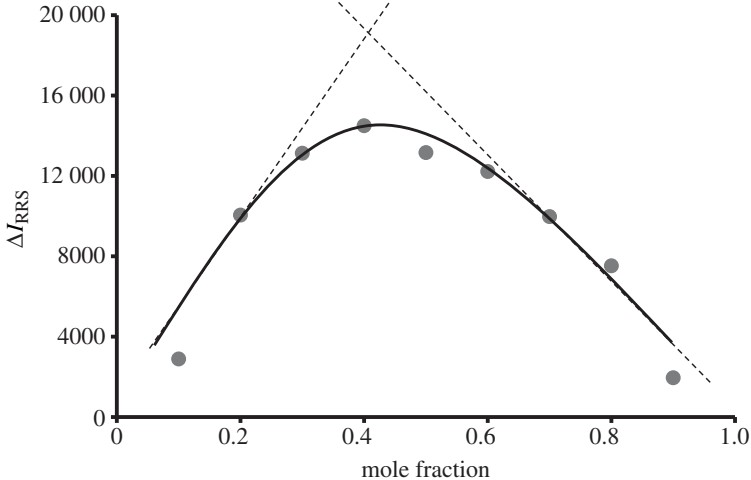

**Figure 4.** Job's plot for the interaction of tebinafine with erythrosine B.

replaced by using the molecular weight, through using the following equation [22]:

$$I = KCMI_\circ,$$

where $I$ refers to the RRS intensity, $K$ refers to a coefficient, $C$ refers to the concentration of the solution, $M$ refers to the molecular weight and $I_\circ$ refers to the incident intensity. When other factors in the equation are fixed, the RRS intensity is directly proportional to the molecular weight ($M$). In the present study, it was found that, increasing the molecular weight from 291.43 (terbinafine$^+$) to 1171.29 (terbinafine-erythrosine B) leads to an increase in the RRS intensity as a result of the interaction between the studied drug and erythrosine B.

### 3.2.3. Formation of a hydrophobic interface

The formation of a hydrophobic interface is directly proportional to the RRS signal [26]. Before the reaction, terbinafine hydrochloride existed as protonated cations while erythrosine B dye existed as anions. Both of them are present in the solutions as water-soluble ions. Accordingly, they can easily form hydrates in water which have very weak RRS intensity. When terbinafine and erythrosine B react with each other, an ion-pair complex (neutralized form) would be formed which leads to the appearance of a hydrophobic liquid–solid interface in the solution. This hydrophobic interface is owing to the presence of the hydrophobic aryl framework in the formed ion-associate binary complex.

### 3.2.4. Effect of the rigidity and molecular planarity

As a result of the binding between the negative and positive charges to form the binary complex, the rotations of aryl group become more restricted and the molecule becomes more rigid and planar. In addition, the molecular volume was increased. These phenomena produce an enhancement in the scattering intensity [27].

### 3.2.5. Determination of stoichiometric ratio of the reaction

Job's method [28] of continuous variation was employed under the working conditions to estimate the stoichiometry of the reaction between the reagent and the drug. Equi-molar solutions of both terbinafine and erythrosine B reagent ($1 \times 10^{-3}$ M) were prepared. Portions of master solutions of the drug and reagent were mixed, comprising different complementary proportions and the general procedure was applied. The RRS intensity of each solution was measured. A blank experiment was carried out simultaneously. Job's plot for the reaction between terbinafine and erythrosine B is shown in figure 4. It is clear from this figure that the ratio is 1 : 1 between the drug and the dye. This result is consistent with the presence of only one basic centre (tertiary amino group) in the drug that is capable for forming an ion-pair complex with the ionized hydroxyl group of the dye as shown in figure 1.

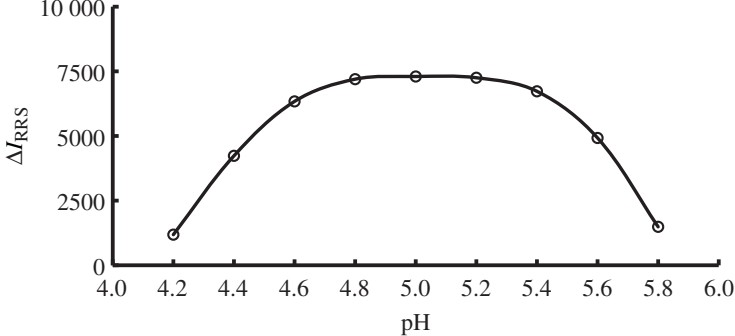

**Figure 5.** Effect of pH on the RRS enhancement of the association complex formation between terbinafine (1.0 µg ml$^{-1}$) and erythrosine B.

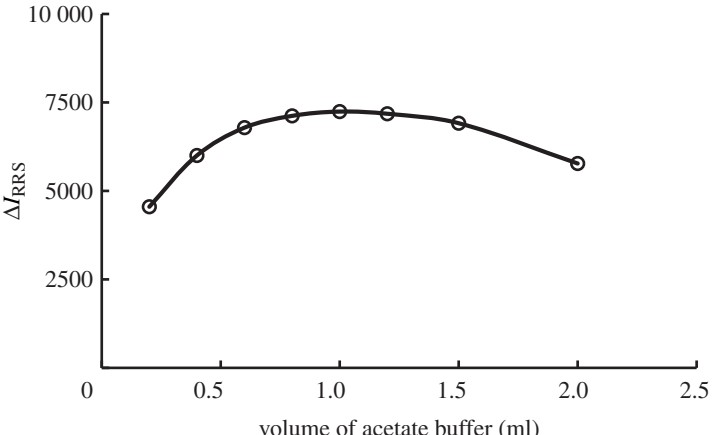

**Figure 6.** Effect of acetate buffer volume (0.2 M) on the RRS enhancement of the association complex formation with terbinafine (1.0 µg ml$^{-1}$).

Furthermore, the binding constant ($K$) between the dye and the drug was calculated according to the modified Benesi–Hildebrand equation [29]:

$$\frac{1}{(I - I_o)} = \frac{1}{(I_{max} - I_o)} + \frac{1}{K[D](I_{max} - I_o)},$$

where $I_p$, $I$ and $I_{max}$ are the Rayleigh scattering intensities of the blank (dye alone), the dye in the presence of the drug and at full saturation, respectively. [$D$] is the molar drug concentration. When $1/(I - I_o)$ was plotted versus $1/[D]$, a linear relationship was obtained. The binding constant ($K$) was calculated from the values of the slope and the intercept ($K$ = intercept/slope), which was found to be $3.03 \times 10^3$.

## 3.3. Optimization of the reaction conditions

### 3.3.1. Influence of acidity and the buffer concentration

The formation of the terbinafine-erythrosine B complex is highly affected by the pH of the reaction medium. Therefore, the reaction was investigated in a pH range of 4.2–5.8 using acetate buffer (0.2 M). In addition, the effect of the 0.2 M acetate buffer volume was investigated. It was found that the maximum RRS intensity ($\Delta I_{RRS}$) values were obtained in the pH range of 4.8–5.2 using 1.0 ml acetate buffer solution. Lower or higher pH value from the mentioned range (4.8–5.2) leads to a marked decrease in the $\Delta I_{RRS}$ intensity of the solution. Therefore, the selected acetate buffer volume was 1.0 ml with pH value of 5.0, figures 5 and 6.

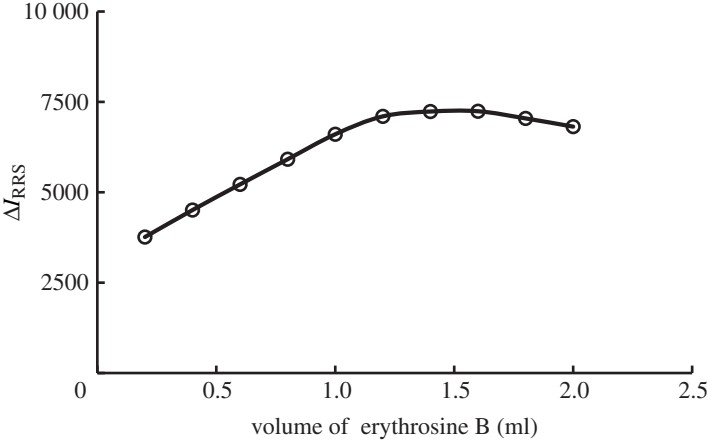

**Figure 7.** Effect of erythrosine B volume ($2.27 \times 10^{-4}$ M) on the RRS enhancement of the association complex formation with terbinafine (1.0 µg ml$^{-1}$).

### 3.3.2. Effect of erythrosine B concentration

The experimental results manifest that the maximum RRS intensity was achieved by using 1.4 ml of erythrosine B solution ($2.27 \times 10^{-4}$ M), figure 7. If erythrosine B concentration was decreased, the reaction would be incomplete. In agreement with the reported data [22], it was found that, concentrations of erythrosine B higher than those mentioned above, leads to self-aggregation of erythrosine B which reduces the observed intensity.

### 3.3.3. Effect of diluting solvents

Various types of diluting solvents were tried to enhance the RRS intensity of the formed binary complex. Low results were obtained with dimethyl sulfoxide and ethanol. The intensity was greatly enhanced with dimethylformamide and acetonitrile. On the other hand, the best results were achieved with distilled water. Therefore, water was selected which is the best green solvent.

## 3.4. Validation of the analytical method

The proposed RRS-based method was validated according to the International Council of Harmonization (ICH) guidelines [30]. The validated parameters were linearity, precision, accuracy, robustness, limit of quantification (LOQ) and limit of detection (LOD). All validation parameters were performed by carrying out the general procedure on standard solutions of terbinafine hydrochloride.

### 3.4.1. Linearity and analytical parameters

A series of standard drug solutions were analysed by the proposed method. By plotting the obtained intensities against the terbinafine hydrochloride concentrations in µg ml$^{-1}$, the proposed method was found to be linear in the concentration range of 0.1–1.5 µg ml$^{-1}$. The statistical parameters for the suggested method were calculated and are presented in table 1. The following equations were applied for the estimation of LOQ and LOD [30]:

$$\mathrm{LOD} = 3.3\,\frac{S_a}{b} \quad \text{and} \quad \mathrm{LOQ} = 10\,\frac{S_a}{b},$$

where $S_a$ refers to the standard deviation of the intercept, and $b$ refers to the slope of the calibration graph. As shown in table 1, the calculated values were 0.089 and 0.029 µg ml$^{-1}$ for LOQ and LOD, respectively.

### 3.4.2. Accuracy, precision and robustness

Different concentrations of the standard terbinafine hydrochloride solution within the range of the calibration curve were analysed by the general analytical procedure. As observed in table 2, the

**Table 1.** Analytical parameters for the determination of terbinafine by the proposed RRS method.

| parameter | value |
|---|---|
| linear range ($\mu g\ ml^{-1}$) | 0.1–1.5 |
| slope | 7260 |
| standard deviation of slope ($S_b$) | 70.4 |
| intercept | −87.4 |
| standard deviation of intercept ($S_a$) | 65 |
| determination coefficient ($r^2$) | 0.9996 |
| correlation coefficient ($r$) | 0.9998 |
| number of determinations | 6 |
| standard deviation of residuals ($S_{y/x}$) | 90.8 |
| limit of quantitation ($\mu g\ ml^{-1}$) | 0.089 |
| limit of detection ($\mu g\ ml^{-1}$) | 0.029 |

**Table 2.** Validation parameters data for the analysis of terbinafine in pure form by the suggested RRS method. (s.d., standard deviation.)

| parameter | $\mu g\ ml^{-1}$ | % recovery ± s.d. |
|---|---|---|
| accuracy[a] | 0.25 | 99.76 ± 1.68 |
|  | 0.75 | 100.68 ± 1.14 |
|  | 1.25 | 100.61 ± 1.40 |
| intra-day precision[a] | 0.25 | 99.76 ± 1.39 |
|  | 0.75 | 100.96 ± 0.96 |
|  | 1.25 | 101.70 ± 0.41 |
| inter-day precision[a] | 0.25 | 101.23 ± 1.44 |
|  | 0.75 | 99.31 ± 1.48 |
|  | 1.25 | 101.39 ± 1.19 |

[a]Mean of three determinations.

accuracy of the proposed method was indicated by the closeness of the resulting values to the true values, while the good precision for the suggested approach was indicated by the low values of the relative standard deviation values for both inter-day and intra-day precision levels, table 2. In addition, the robustness of the proposed approach indicated by the consistency of the RRS intensity with the minor changes in optimal parameters such as changes in pH (5.0 ± 0.1), changes in erythrosine B volume (1.4 ± 0.1 ml) and changes in acetate buffer volume (1.0 ± 0.1 ml). It was found that the RRS intensity did not significantly affect any of these minor changes which prove the robustness of the proposed approach (table 3).

## 3.5. Determination of terbinafine in pharmaceutical samples

Terbinafine hydrochloride was successfully determined in different pharmaceutical dosage forms (Lamifen® 1% cream and tablets Lamifen® 125 mg tablets). It was found that the recoveries obtained by the proposed method for terbinafine determination in different pharmaceutical dosage forms are in good agreement with those obtained by analysing the same dosage forms with the reported spectrofluorimetric method [31]. As shown in table 4, the calculated values of the variance ratio F-test, and t-test were less than the tabulated values, indicating that there was no significant difference between the proposed RRS method and the reported spectrofluorimetric method.

**Table 3.** Robustness study of the proposed RRS method for determination of terbinafine in pure form. (s.d., standard deviation.)

| parameter | | % recovery ± s.d.[a] |
|---|---|---|
| optimum | | 101.07 ± 0.96 |
| pH of solution | 4.9 | 98.63 ± 0.48 |
| | 5.1 | 99.0 ± 0.96 |
| volume of erythrosine B solution | 1.5 ml | 100.93 ± 1.76 |
| | 1.3 ml | 98.45 ± 1.68 |
| volume of buffer solution | 1.1 ml | 100.29 ± 1.39 |
| | 0.9 ml | 101.75 ± 0.69 |

[a]Mean of three replicate measurement of 1.0 µg ml$^{-1}$ terbinafine.

**Table 4.** Estimation of terbinafine in pharmaceutical dosage forms by the suggested RRS and the reported spectrofluorimetric methods.

| dosage form | found terbinafine | | *t*-value[b] | *F*-value[b] |
|---|---|---|---|---|
| | proposed method | reported method | | |
| Lamifen® 125 mg tablets | 127.81 mg tablet$^{-1}$ | 124.28 mg tablet$^{-1}$ | 1.99 | 3.2 |
| Lamifen® %cream | 1.02% w/w | 0.993% w/w | 1.83 | 2.09 |

[a]Average of five determinations.
[b]Tabulated value at 95% confidence limit; $F = 6.338$ and $t = 2.306$.

# 4. Conclusion

In a slightly acidic medium, terbinafine hydrochloride can react with erythrosine B through hydrophobic forces and electrostatic attraction to form an ion-pair complex. Depending on this reaction, a novel spectral RRS methodology was proposed and validated for the estimation of terbinafine hydrochloride with good selectivity, high sensitivity, simplicity and speed. In addition, the presented work is environmentally friendly as no organic solvent is used in the analytical procedure. In addition, the validity of the method was investigated according to ICH guidelines. The proposed procedure was successfully applied to determine terbinafine hydrochloride in different pharmaceutical tablets, and cream samples with sufficient recovery and additionally the accuracy and precision were confirmed by Student's *t*-test and variance ratio *F*-test.

Data accessibility. Data are available at Dryad Digital Repository, https://doi.org/10.5061/dryad.9s4mw6mdm [32].
Authors' contributions. M.A.A.-L. carried out the laboratory work, participated in data analysis and participated in the design of the study. S.M.D. designed the study, coordinated the study and submitted the manuscript. D.A.M.N.E.-D. participated in data analysis, participated in the design of the study and drafted the manuscript. A.A. carried out some experimental work and revised the manuscript. M.O. carried out the statistical analysis, conceived of the study and drafted the manuscript. All authors gave final approval for publication.
Competing interests. There is no conflict of interest to declare.
Funding. We received no funding for this study.

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
