## [Reviewer comments · Royal Society Open Science]

Review History

RSOS-201545.R0 (Original submission)

Review form: Reviewer 1 (Keith Gordon)

Is the manuscript scientifically sound in its present form?

Yes

Are the interpretations and conclusions justified by the results?

No

Is the language acceptable?

Yes

Do you have any ethical concerns with this paper?

No

Have you any concerns about statistical analyses in this paper?

No

Recommendation?

Major revision is needed (please make suggestions in comments)

Comments to the Author(s)

See attached document (Appendix A).

Review form: Reviewer 2**Is the manuscript scientifically sound in its present form?**

Yes

Are the interpretations and conclusions justified by the results?

Yes

Is the language acceptable?

Yes

Do you have any ethical concerns with this paper?

No

Have you any concerns about statistical analyses in this paper?

No

Recommendation?

Accept with minor revision (please list in comments)

Comments to the Author(s)

In this manuscript Abdel-Lateef and co-workers investigate of terbinafine with erythrosine B or its determination in different pharmaceutical formulations based on resonance Rayleigh scattering technique. The overall quality and novelty of the report are not sufficient to publish the current version of the manuscript. Minor revision is required before its publication.

1. There are multiple typos and require proof-reading
2. The operational conditions of spectrofluorimeter such as voltage, lamp and etc. should be written.
3. Section 2.2., United kingdom and city should be corrected. The concentration of reactants should be same unit.
4. Section 2.3, units must be used similar way, mg/mL mg mL⁻¹, ng mL⁻¹ one of them must be chosen as a rules of journal
5. The equations must be given separately from the text. Symbols in equation and text must be same.
6. In figure 5, the effect of dye volumes was tried at 1 μ g /mL terbinafine concentration which was a medium concentration of calibration. Why did not try at the max concentration? Maybe it will be needed more volume of dye.
7. Linear range was written as 1.0 - 15.0 μ g /mL in section 2.3, and 0.1-1.5 μ g /mL in others section. Which one is true?
8. In Table 3, for robustness studies Volume of erythrosine B solution was selected as 1.1 and 0.9 mL. But in general procedure author claimed that erythrosine B was used as 1.4 mL. Table have be corrected. Similarly, acetate buffer volumes also should be corrected. Subtitle of Table 3, written concentration is for terbinafine, isn't it?
9. For Table 4, it is better to give as a tablet or cream quantity instead of recovery values.
10. Sub titles of tables should be written small font.

Review form: Reviewer 3

Is the manuscript scientifically sound in its present form?

Yes

Are the interpretations and conclusions justified by the results?

Yes

Is the language acceptable?

Yes

Do you have any ethical concerns with this paper?

No

Have you any concerns about statistical analyses in this paper?

No

Recommendation?

Major revision is needed (please make suggestions in comments)

Comments to the Author(s)

The manuscript is clear but I have some points:

- 1- why the minimum intensity is 500? Can't be lower than this?
- 2- what is the advantage of this method regarding that terbinafine has been determined based on its native fluorescence without derivatization. The method could be more useful for a non fluorescent drug.
- 3- I think you can provide a comparison for all spectrofluorimetric methods for the drugs
- 4- why you didn't make a molar ratio by practical method?
- 5- The slope is very high please revise

Review form: Reviewer 4

Is the manuscript scientifically sound in its present form?

Yes

Are the interpretations and conclusions justified by the results?

Yes

Is the language acceptable?

Yes

Do you have any ethical concerns with this paper?

No

Have you any concerns about statistical analyses in this paper?

No

Recommendation?

Accept with minor revision (please list in comments)

Comments to the Author(s)

Referee: #

Dear Prof. Dr. Ya-Wen Wang

Referring to manuscript ID: RSOS-201545; entitled "Investigating the interaction of terbinafine with xanthenes dye for its feasible determination applying resonance Rayleigh scattering technique"

Recommended: minor revision

1- The authors should clarify the novelty of the proposed method over the published method by; F. Belal; which has lower LOD and LOQ, 3.0 and 9.0 ng/mL for terbinafine respectively.

DOI: 10.1007/s10895-013-1237-3

2- Conditional stability constant (Kf) of the ion-pair complex need to be evaluate.

3- Stoichiometric relationship between (drug: dye) did not postulate?

4- Author stated that order of addition has an effect, despite the reaction is being electrostatic attraction to form association complex, Explain?

Decision letter (RSOS-201545.R0)

Dear Professor Derayea:

Title: Investigating the interaction of terbinafine with xanthenes dye for its feasible determination applying resonance Rayleigh scattering technique

Manuscript ID: RSOS-201545

The editor assigned to your manuscript has now received comments from reviewers. We would like you to revise your paper in accordance with the referee and Subject Editor suggestions which can be found below (not including confidential reports to the Editor). Please note this decision does not guarantee eventual acceptance.

Please submit your revised paper before 07-Nov-2020. Please note that the revision deadline will expire at 00.00am on this date. If we do not hear from you within this time then it will be assumed that the paper has been withdrawn. In exceptional circumstances, extensions may be possible if agreed with the Editorial Office in advance. We do not allow multiple rounds of revision so we urge you to make every effort to fully address all of the comments at this stage. If deemed necessary by the Editors, your manuscript will be sent back to one or more of the original reviewers for assessment. If the original reviewers are not available we may invite new reviewers.

When submitting your revised manuscript, you must respond to the comments made by the referees and upload a file "Response to Referees" in "Section 6 - File Upload". Please use this to

document how you have responded to the comments, and the adjustments you have made. In order to expedite the processing of the revised manuscript, please be as specific as possible in your response.

RSC Associate Editor:
Comments to the Author:
(There are no comments.)

RSC Subject Editor:
Comments to the Author:
(There are no comments.)

Reviewers' Comments to Author:
Reviewer: 1

Comments to the Author(s)
See attached document

Reviewer: 2

Comments to the Author(s)
In this manuscript Abdel-Lateef and co-workers investigate of terbinafine with erythrosine B or its determination in different pharmaceutical formulations based on resonance Rayleigh scattering technique. The overall quality and novelty of the report are not sufficient to publish the current version of the manuscript. Minor revision is required before its publication.

1. There are multiple typos and require proof-reading
2. The operational conditions of spectrofluorimeter such as voltage, lamp and etc. should be written.
3. Section 2.2., United kingdom and city should be corrected. The concentration of reactants should be same unit.
4. Section 2.3, units must be used similar way, mg/mL mg mL⁻¹, ng mL⁻¹ one of them must be chosen as a rules of journal
5. The equations must be given separately from the text. Symbols in equation and text must be same.

6. In figure 5, the effect of dye volumes was tried at 1 $\mu\text{g}/\text{mL}$ terbinafine concentration which was a medium concentration of calibration. Why did not try at the max concentration? Maybe it will be needed more volume of dye.
7. Linear range was written as 1.0 - 15.0 $\mu\text{g}/\text{mL}$ in section 2.3, and 0.1-1.5 $\mu\text{g}/\text{mL}$ in others section. Which one is true?
8. In Table 3, for robustness studies Volume of erythrosine B solution was selected as 1.1 and 0.9 mL. But in general procedure author claimed that erythrosine B was used as 1.4 mL. Table have be corrected. Similarly, acetate buffer volumes also should be corrected. Subtitle of Table 3, written concentration is for terbinafine, isn't it?
9. For Table 4, it is better to give as a tablet or cream quantity instead of recovery values.
10. Sub titles of tables should be written small font.

Reviewer: 3

Comments to the Author(s)

The manuscript is clear but I have some points:

- 1- why the minimum intensity is 500? Can't be lower than this?
- 2- what is the advantage of this method regarding that terbinafine has been determined based on its native fluorescence without derivatization. The method could be more useful for a non fluorescent drug.
- 3- I think you can provide a comparison for all spectrofluorimetric methods for the drugs
- 4- why you didn't make a molar ratio by practical method?
- 5- The slope is very high please revise

Reviewer: 4

Comments to the Author(s)

Referee: #

Dear Prof. Dr. Ya-Wen Wang

Referring to manuscript ID: RSOS-201545; entitled "Investigating the interaction of terbinafine with xanthenes dye for its feasible determination applying resonance Rayleigh scattering technique"

Recommended: minor revision

- 1- The authors should clarify the novelty of the proposed method over the published method by; F. Belal; which has lower LOD and LOQ, 3.0 and 9.0 ng/mL for terbinafine respectively.
DOI: 10.1007/s10895-013-1237-3
- 2- Conditional stability constant (Kf) of the ion-pair complex need to be evaluate.
- 3- Stoichiometric relationship between (drug: dye) did not postulate?
- 4- Author stated that order of addition has an effect, despite the reaction is being electrostatic attraction to form association complex, Explain?

Author's Response to Decision Letter for (RSOS-201545.R0)

See Appendix B.

RSOS-201545.R1 (Revision)

Review form: Reviewer 2

Is the manuscript scientifically sound in its present form?

Yes

Are the interpretations and conclusions justified by the results?

Yes

Is the language acceptable?

Yes

Do you have any ethical concerns with this paper?

No

Have you any concerns about statistical analyses in this paper?

No

Recommendation?

Accept as is

Comments to the Author(s)

All corrections were made according to the reviewers suggestions. As a result, the manuscript can be acceptable as is

Review form: Reviewer 3

Is the manuscript scientifically sound in its present form?

Yes

Are the interpretations and conclusions justified by the results?

Yes

Is the language acceptable?

Yes

Do you have any ethical concerns with this paper?

No

Have you any concerns about statistical analyses in this paper?

No

Recommendation?

Accept as is

Comments to the Author(s)

No further comments

Review form: Reviewer 4

Is the manuscript scientifically sound in its present form?

Yes

Are the interpretations and conclusions justified by the results?

Yes

Is the language acceptable?

Yes

Do you have any ethical concerns with this paper?

No

Have you any concerns about statistical analyses in this paper?

No

Recommendation?

Accept as is

Comments to the Author(s)

Accepted

Decision letter (RSOS-201545.R1)

Dear Professor Derayea:

Title: Investigating the interaction of terbinafine with xanthenes dye for its feasible determination applying resonance Rayleigh scattering technique
Manuscript ID: RSOS-201545.R1

It is a pleasure to accept your manuscript in its current form for publication in Royal Society Open Science. The chemistry content of Royal Society Open Science is published in collaboration with the Royal Society of Chemistry.

Royal Society of Chemistry
Thomas Graham House
Science Park, Milton Road

Cambridge, CB4 0WF
Royal Society Open Science - Chemistry Editorial Office

RSC Associate Editor:
Comments to the Author:
(There are no comments.)

RSC Subject Editor:
Comments to the Author:
(There are no comments.)

Reviewer(s)' Comments to Author:
Reviewer: 2

Comments to the Author(s)
All corrections were made according to the reviewers suggestions. As a result, the manuscript can be acceptable as is

Reviewer: 3

Comments to the Author(s)
No further comments

Reviewer: 4

Comments to the Author(s)
Accepted

Appendix A

Review of RSOS-201545

This paper reports the use of resonance Rayleigh scattered to investigate how an ion paired complex may be used in the evaluation of drug levels in a formulation of terbinafine hydrochloride.

There are a couple of issues that are problematic I think, these include:

- Resonance Rayleigh scattering is not routine yet there is no real description of what is actually being done here. I think it is required. Papers such as Resonance Rayleigh scattering of cyanine dyes in solution by J. Anglister and I. Z. Steinberg *J. Chem. Phys.* 78, 5358 (1983); <https://doi.org/10.1063/1.445489> are good resources as is *Anal. Chem.* 2001, 73, 3907-3914.
- If one reads the Steinberg paper it becomes evident that the UV-vis absorption spectrum is related to the resonance Rayleigh response. Would it not be a good idea to show the UV-vis spectrum of the ion-pair complex?
- The statement "When fixation other factors in the equation, the resonance Rayleigh scattering intensity is directly proportional to the particle molecular weight (M)." seems odd to me. A critical issue in scattering with resonance effects is ϵ^2 ; that is the absorption intensity squared. So the argument that the mass of the dye is boosting the signal is surely a secondary effect to the presence of an intense absorption due to the ion-pair formation? I realize that is stated in the *Talanta* paper that is cited – I just think it is a secondary issue.
- The sample? What actually is it? It seems from my reading that it is formulated drug. If this is the case then what has been done about the excipients? Have they even been considered?
- I think the issue of the formulation really throws the paper a bit - in that as the authors do not characterise the formulation or its components there is a danger of calibration data not being transferable. I think this needs fully addressed - perhaps use the API alone.
- Finally, this is not such a novel paper - it does rather closely follow the *Talanta* paper referenced as ref 21. What justification do the authors give for novelty here?

I think the paper needs major revision

Appendix B

Dear Dr Laura Smith

Publishing Editor, Journals

Royal Society of Chemistry

Thank you for giving us the opportunity to further revise our Manuscript **Ref. ID RSOS-201545** entitled "**Investigating the interaction of terbinafine with xanthenes dye for its feasible determination applying resonance Rayleigh scattering technique**" for consideration to be published in Royal Society Open Science Journal.

We have carefully revised our manuscript according to the reviewer`s comments. We are grateful for the comments and advice that we have received. We hope that we have answered all comments satisfactorily.

N.B. All corrections have been re-written in red color in the revised manuscript.

Sincerely,

Sayed Derayea

Reviewer #1:

1- "Resonance Rayleigh scattering is not routine yet there is no real description of what is actually being done here. I think it is required. Papers such as Resonance Rayleigh scattering of cyanine dyes in solution by J. Anglister and I. Z. Steinberg J. Chem. Phys. 78, 5358 (1983); <https://doi.org/10.1063/1.445489> are good resources as is Anal. Chem. 2001, 73, 3907-3914.

The suggested articles were cited in the appropriate locations in the manuscript.

2-"If one reads the Steinberg paper it becomes evident that the UV-vis absorption spectrum is related to the resonance Rayleigh response. Would it not be a good idea to show the UV-vis spectrum of the ion-pair complex?"

The UV-Vis spectrum of the ion-pair complex was included in the manuscript and was assigned as Figure 3.

3- The statement "When fixation other factors in the equation, the resonance Rayleigh scattering intensity is directly proportional to the particle molecular weight (M)." seems odd to me. A critical issue in scattering with resonance effects is ϵ^2 ; that is the absorption intensity squared. So the argument that the mass of the dye is boosting the signal is surely a secondary effect to the presence of an intense absorption due to the ion pair formation? I realize that is stated in the Talanta paper that is cited - I just think it is a secondary issue."

The molecular weight is one factor among other that affect resonance Rayleigh scattering. There are many published article that support the enhancement effect of the molecular weight on the intensity of RRS other than the article that was published by Talanta, such as:-

[a] X. Wei, Z. Liu, S. Liu, Resonance Rayleigh scattering spectra of tetracycline antibiotic-Cu (II)-titan yellow systems and their applications in analytical chemistry, *Analytical Bioanalytical Chemistry* 385(6) (2006) 1039-1044.

[b] F. Tian, W. Huang, J. Yang, Q. Li, Study on the interaction between albendazole and eosin Y by fluorescence, resonance Rayleigh scattering and frequency doubling scattering spectra and their analytical applications, *Spectrochimica Acta Part A: Molecular Biomolecular Spectroscopy* 126 (2014) 135-141.

[c] X. Liu, Z. Zhang, J. Peng, Y. He, High-performance liquid chromatography with resonance Rayleigh scattering detection for determining four tetracycline antibiotics, *Analytical Methods* 6(23) (2014) 9361-9366.

4- The sample? What actually is it? It seems from my reading that it is formulated drug. If this is the case then what has been done about the excipients? Have they even been considered?

Yes, the sample means formulated drug (tablets). Lamisil® and Lamifen® tablets contain the following excipients: magnesium stearate, colloidal anhydrous silica,

hydroxy propyl methylcellulose, sodium carboxy methyl starch, microcrystalline cellulose. All of them do not contain any basic center (secondary, tertiary or quaternary amines) in their chemical structure, accordingly these compounds are not able to form ion pair complex can be formed with the acidic xanthene dye. This can be evidenced from the results presented in Table 4.

5- I think the issue of the formulation really throws the paper a bit - in that as the authors do not characterise the formulation or its components there is a danger of calibration data not being transferable. I think this needs fully addressed - perhaps use the API alone.

The method was applied for the determination of the cited drug in its pharmaceutical preparations (tablets and cream). The source and description of these dosage forms were illustrated at the end of second "2.2 Materials and reagent" in the experimental part. In addition the preparation of these formulation for analysis was described in second "2.4. Procedures for pharmaceutical tablets " and "2.5. Procedures for pharmaceutical cream " in the experimental part. It should be mentioned here that, the full composition of each dosage form is not usually included in the published papers.

However, the validation was performed using standard terbinafine hydrochloride solution." As mentioned under the section "3.4. Validation the analytical method " .

6- Finally, this is not such a novel paper - it does rather closely follow the Talanta paper referenced as ref 21. What justification do the authors give for novelty here?

The Talanta paper was reported for the determination of other drug "propranolol" but the current paper was designated for terbinafine. The novelty of the present work was described at end of the "1. Introduction" of the manuscript.

Reviewer #2:

1. " There are multiple typos and require proof-reading."

The manuscript was deeply revised.

2. " The operational conditions of spectrofluorimeter such as voltage, lamp and etc. should be written."

The required data were written in the manuscript and highlighted with red color.

3."Section 2.2., United kingdom and city should be corrected. The concentration of reactants should be same unit."

Done

4. "Section 2.3, units must be used similar way, mg/mL mg mL⁻¹, ng mL⁻¹ one of them must be chosen as a rules of journal"

The unit was unified.

5. "The equations must be given separately from the text. Symbols in equation and text must be same."

Done

6. In figure 5, the effect of dye volumes was tried at 1 µg /mL terbinafine concentration which was a medium concentration of calibration. Why did not try at the max concentration? Maybe it will be needed more volume of dye.

The drug reacts with the dye in a molar ratio of 1:1 and the used concentration of the dye was far above the upper concentration limit of the linear range. Furthermore, the proposed method was validated at three concentration levels, one of them is 1.25 µg /mL. In addition the analysis for calibration curve construction was carried out in the concentration range of 0.1-1.5 µg /mL and the correlation coefficient was excellent. Therefore, we think that the concentration of the dye is sufficient for all drug concentrations including 1.5 µg /mL terbinafine.

7. Linear range was written as 1.0 - 15.0 µg /mL in section 2.3, and 0.1-1.5 µg/mL in others section. Which one is true?

In section 2.3 " General analytical procedure" the given concentration is for the original solution (1.0 - 15.0 µg /mL) which were diluted to 10-mL to give final concentration of 0.1 – 1.5 µg /mL terbinafine. Therefore the final concentration range is 0.1 – 1.5 µg /mL. However, the sentence in section 2.3 was rephrased to remove the confusion.

8. In Table 3, for robustness studies Volume of erythrosine B solution was selected as 1.1 and 0.9 mL. But in general procedure author claimed that erythrosine B was used as 1.4 mL. Table have be corrected. Similarly, acetate buffer volumes also should be corrected. Subtitle of Table 3, written concentration is for terbinafine, isn't' it?

All written errors in Table 3 have been corrected.

9. For Table 4, it is better to give as a tablet or cream quantity instead of recovery values.

Done

10. Sub titles of tables should be written small font.

Done

Reviewer #3:

1- why the minimum intensity is 500? Can't be lower than this?

The resonance Rayleigh scattering intensity of the blank (xanthenes dye) was 500. Therefore, we could not use values lower than that of the blank.

2- what is the advantage of this method regarding that terbinafine has been determined based on its native fluorescence without derivatization. The method could be more useful for a non fluorescent drug.

The proposed method characterized by a wide linear range (0.1 – 1.5 µg /mL) which allow to broad applications for the determination of terbinafine. While the reported spectrofluorimetric method (DOI: 10.1007/s10895-013-1237-3) has a narrow linear range of 0.02-0.15 µg /mL

3- I think you can provide a comparison for all spectrofluorimetric methods for the drugs.

Only one spectrofluorimetric method was reported for the determination of terbinafine. DOI: 10.1007/s10895-013-1237-3

4- why you didn't make a molar ratio by practical method?

The a molar ratio for the reaction was practically performed and added in the manuscript under section "3.2.5. Determination of stoichiometric ratio of the reaction"

5- The slope is very high please revise

The values of the obtained values of resonance Rayleigh scattering are very high up to 20.000. The spectrometer can measure reading up to 50.000 relative intensity. Consequently the value of the slope is 7260 when the used terbinafine concentration is in µg/mL. However, if the terbinafine concentration is in ng/mL, the obtained slope value would be 7.2 .

Reviewer #4:

1- The authors should clarify the novelty of the proposed method over the published method by; F. Belal; which has lower LOD and LOQ, 3.0 and 9.0 ng/mL for terbinafine respectively.

DOI: 10.1007/s10895-013-1237-3

The proposed method characterized by a wide linear range (0.1 – 1.5 µg /mL) which allow to broad applications for the determination of terbinafine. While the

reported spectrofluorimetric method has a narrow linear range of 0.02-0.15 μg /mL

2- Conditional stability constant (K_f) of the ion-pair complex need to be evaluate.

The binding constant between the drug and the dye was estimated by applying modified Benesi–Hildebrand equation.

3- Stoichiometric relationship between (drug: dye) did not postulate?

The molar ratio for the reaction was practically performed and included in the manuscript under section "3.2.5. Determination of stoichiometric ratio of the reaction"

4- Author stated that order of addition has an effect, despite the reaction is being electrostatic attraction to form association complex, Explain?

The mentioned order of addition gives slight improvement in the obtained results. This could be explained as follow; Terbinafine hydrochloride should be protonated firstly by adding acidic buffer solution to facilitate the binary complex formation with the xanthene dyes. However, the effect of the order of addition is not critical so this factor was removed from the manuscript..